# Lnc-uc.147 Is Associated with Disease Stage of Liver, Gastric, and Renal Cancer

**DOI:** 10.3390/biom13020265

**Published:** 2023-01-31

**Authors:** Ana Carolina Rodrigues, Erika Pereira Zambalde, Daniel de Lima Bellan, Edvaldo da Silva Trindade, Enilze Maria de Souza Fonseca Ribeiro, George Calin, Daniela Fiori Gradia, Jaqueline Carvalho de Oliveira

**Affiliations:** 1Laboratory of Human Cytogenetics and Oncogenetics, Department of Genetics, Universidade Federal do Paraná, Curitiba 19031, PR, Brazil; 2Multidisciplinary Laboratory of Food and Health, School of Applied Sciences, State University of Campinas, Limeira 13484-350, SP, Brazil; 3Department of Cell Biology, Universidade Federal do Paraná, Curitiba 19031, PR, Brazil; 4Department of Experimental Therapeutics, The University of Texas MD Anderson Cancer Center, Houston, TX 77030, USA; 5Center for RNA Interference and Non-Coding RNAs, The University of Texas MD Anderson Cancer Center, Houston, TX 77030, USA

**Keywords:** lnc-uc.147, uc.147, T-UCRs, liver cancer, lncRNAs, TEA domain transcription factor 4

## Abstract

Lnc-uc.147, a long non-coding RNA derived from a transcribed ultraconserved region (T-UCR), was previously evidenced in breast cancer. However, the role of this region in other tumor types was not previously investigated. The present study aimed to investigate lnc-uc.147 in different types of cancer, as well as to suggest lnc-uc.147 functional and regulation aspects. From solid tumor datasets analysis of The Cancer Genome Atlas (TCGA), deregulated lnc-uc.147 expression was associated with the histologic grade of hepatocellular carcinoma, and with the tumor stage of clear cell renal and gastric adenocarcinoma. Considering the epidemiologic relevance of liver cancer, silencing lnc-uc.147 reduced the viability and clonogenic capacity of HepG2 cell lines. Additionally, we suggest a relation between the transcription factor TEAD4 and lnc-uc.147 in liver and breast cancer cells.

## 1. Introduction

Ultraconserved regions (UCRs) make up a total of 481 DNA segments that are 100% conserved in human, mouse, and rat genomes, with also high conservation in the orthologous regions of fungus, chicken, dog, and other species [1,2]. In 2007, Calin et al. [3] showed that these regions are transcribed in different tissues, named transcribed UCRs (T-UCRs). The high level of conservation in these molecules suggests an important cell role. In this context, T-UCRs are being studied in different physiologic and pathologic processes, such as cancer, ischemia, apoptosis, thermogenesis, intestinal barrier renewal, and other conditions [4]. 

In a previous study from our group, a uc.147 transcript was characterized and studied in breast cancer (BC). The long non-coding RNA (lncRNA) from uc.147 (lnc-uc.147) is located in the nucleus and has approximately 2.8 kb. The increased expression of lnc-uc.147 was associated with the luminal A and B subtypes of BC, also leading to poor prognosis in those groups. Furthermore, silencing the transcript increased apoptosis, arrested the cell cycle, and reduced colony formation and cell viability in BC cell lines [5]. As far as we know, this is the only study associating this lncRNA with cancer. 

Here, we aimed to identify the role of lnc-uc.147 in other tumor types and to understand the mechanisms of regulation in this lncRNA.

## 2. Materials and Methods

### 2.1. In Silico Analysis

For lncRNA expression analysis, the interactive platform TANRIC (The Atlas of Non-coding RNA in Cancer) was used [6]. LncRNA expression data were analyzed in 20 tumor types, and normal samples from patients were included in the TCGA database. For the analysis of the uc.147 region on this platform, the chromosome’s position of this region was used (chr4:151236383-151236690—GRCh37). For transcription factor binding site analysis, the TFBSPred online tool was used [7]. The chromosome location used in this tool was 4:150315231(+)—GRCh38. 

### 2.2. Cell Culture and Growth Conditions

HepG2, a human liver cancer cell line, as well as BT-474 and CAMA-1, both luminal breast cancer cell lines, were cultured in Dulbecco’s modified Eagle’s medium (DMEM) (Gibco, Waltham, MA, USA), supplemented with 10% FBS (Cultilab, Campinas, Brazil) and penicillin–streptomycin (Gibco), 100 U/mL and 100 µg/mL, respectively. All cells were placed at 37 °C with a humidified atmosphere of 5% CO_2_.

### 2.3. RNA Isolation, cDNA Synthesis, and Quantitative Real-Time PCR Analysis

The total RNA from transfected cells was extracted using TRIzol reagent (Invitrogen, Waltham, MA, USA), following the manufacturer’s protocol. The concentration of the RNA was assessed using the NanoDrop 2000 spectrophotometer. cDNA was synthesized using the Invitrogen SuperScript III Reverse Transcriptase kit (Invitrogen), and 10-times diluted cDNA was used for qPCR analysis using the PowerUp^TM^ SYBR Green Master Mix (Applied Biosystems, Waltham, MA, USA) with the appropriate primers (Table 1), and the Viia7 System equipment was utilized. A negative RT control (-RT) for each sample was used to confirm the DNase treatment (Thermo Scientific™, Waltham, MA, USA, Catalog number: EN0525). The 2^−ΔΔCt^ method was used to calculate the relative abundance of RNA genes compared with two of the following genes: GAPDH, U6, and TBP expression. All reactions were performed at least in duplicate, with negative control, and a maximum standard deviation of 0.5 was accepted. The primer sequences are presented in Table 1. 

### 2.4. siRNA Treatment

The siRNAs targeting the uc.147 transcript and the transcription factor TEAD4 (NM_003213; siRNA ID: SASI_Hs01_00046458, sequence start: 1490) were purchased from Sigma Aldrich. The sequence of the siRNA uc.147 was 5′-CCAGCAACCAGCAAGUGAAdTdT-3′. Negative control was utilized (siRNA-NC). For siRNA uc.147, a concentration of 100 nM was used, and for siRNA TEAD4, the concentration was 20 nM. The transfection procedures were performed using Lipofectamine^TM^ 3000 (Invitrogen) according to the manufacturer’s protocol. Cells were used for the functional assays and a collection of RNA was used after 24 h of transfection. 

### 2.5. Cell Viability Assay

Cell viability was determined as previously described [8]. For this assay, using the compound resazurin, mitochondrial viability was assessed. Transiently transfected cells were seeded into 96-well plates at a density of approximately 2000 cells per well, and after treatment, resazurin dye (10 µmol/L) was added (Sigma-Aldrich, St. Louis, MO, USA) to each well. Cell viability was tested every 24 h (2, 3, and 4 days after transfection). The plates were incubated for 4 h at 37 °C, and the product was measured at 570 and 600 nm using Biotek Synergy LX Multimode Reader. Each sample was tested in five replicate wells, and three independent experiments were performed.

### 2.6. Cell Proliferation Assay

Cell proliferation was performed based on [9] with modifications. For this assay, crystal violet, a nuclear dye, was used to infer the number of cells in the well after transfection. Cells were seeded into 96-well plates at a density of approximately 6000 cells per well. At 24 and 48 h after transfection, the cells were washed, fixed with methanol 100%, and then stained with 1% violet crystal for 15 min. Then, the cells were washed with PBS, and the dye was solubilized in a solution with 33% glacial acetic acid. The product was measured at 570 using Biotek Synergy LX Multimode Reader. Absorbance was compared with a standard curve previously prepared to determine the number of cells. Each sample was tested in five replicate wells, and three independent experiments were performed.

### 2.7. Colony Formation Assay

For the colony formation assay, 500 cells were seeded into each well of a six-well plate and maintained in a medium containing 10% FBS in an incubator with 5% CO_2_ at 37 °C after incubation with siRNA uc.147 and siRNA-NC. After 7 days, HepG2 cells were washed with PBS, fixed with 100% methanol at room temperature for 15 min, stained with 1% crystal violet for 15 min, and washed with PBS until the excess dye was removed. The number of colonies in each well was counted. Each sample was tested in triplicate wells, and three independent experiments were performed.

### 2.8. Apoptosis and Cell Cycle

The apoptosis ratio was determined using the FITC Annexin V/Dead Cell Apoptosis Kit (Life Technologies). After 48 h of transfection, cells were harvested and resuspended in a binding buffer containing Annexin V-FITC and propidium iodide (PI), according to the manufacturer’s protocol. Samples were analyzed by flow cytometry (BD FACS Calibur). Cells were discriminated into viable, necrotic, and apoptotic cells, and then the percentages of apoptotic cells from each group were compared. Three independent experiments were conducted. For cell cycle analysis, the cells were collected 48 h after treatment with siRNA-NC and siRNA uc.147. They were fixed in 70% ethanol and stored at 4 °C for at least 24 h. Afterward, the cells were stained in a solution containing 50 µg/mL of PI (Life Technologies), 50 µg/mL of RNase A (Invitrogen), and 0.01% Triton-X. Samples were analyzed by flow cytometry (BD FACS Calibur). FlowJo software was used to analyze cell cycle phases, and three independent experiments were performed.

### 2.9. Scratch Assay

For this experiment, 48 h after transfection, cells were plated in 96-well plates. After the formation of a monolayer, scratches were made in the center of the wells with the aid of a plastic tip (P10). Subsequently, the wells were washed twice with serum-free DMEM, which was maintained. Images of all wells were captured after 2 h, 4 h, 8 h, 24 h, and 48 h at 100× magnification. The photos were analyzed in ImageJ and Tscratch programs, determining the percentage of closing in the area over time. Each sample was tested in triplicate wells, and three independent experiments were performed.

### 2.10. Statistical Analysis 

Statistical analysis was performed using GraphPad Prism software (GraphPad Software Inc., La Jolla, CA, USA). Data from three independent experiments were expressed as mean ± standard deviation (SD). Statistical differences were evaluated using the Student *t*-test. A *p*-value of <0.05 was considered statistically significant.

## 3. Results

We evaluated lnc-uc.147 expression in 20 tumor types using TCGA RNA sequencing data through the TANRIC platform. Based on this analysis, we found an association of lnc-uc.147 expression with the histologic grade of hepatocellular carcinoma (*p* = 0.018), the disease stage of stomach adenocarcinoma (*p* = 0.023), and clear renal cell carcinoma (*p* = 0.049), and, as previously described, with breast cancer (Table 2). The graphs generated by the TANRIC platform are represented in Appendix A.

Liver cancer represents the seventh most commonly diagnosed cancer, and the fourth cause of death related to cancer globally [10]. Hepatocellular carcinoma (HCC) is the most frequent subtype of liver cancer, corresponding to approximately 75% of cases [11,12]. Although the studies and clinical trials involving new therapies for this tumor have been growing recently, the mechanisms underlying this disease are still underexplored. In addition, considering the high lethality of this disease, the identification of new biomarkers for prognostic and targets for therapy are needed. Based on this, we focused on studying this tumor.

Through TCGA data analysis, we found that lnc-uc.147 expression was associated with the histological grade of HCC samples. However, no differential expression was found when comparing tumor and non-tumor samples (*p* = 0.35, Appendix A). Based on these results, we suggest that the differential expression of lnc-uc.147 is observed inside the tumor group, and a higher expression of this lncRNA is associated with a higher tumor grade. 

To analyze the phenotypic effects of the lnc-uc.147 knockdown, a siRNA-based approach was applied in the hepatocellular carcinoma cell line, HepG2. The mRNA levels of the host gene *LRBA* were not affected by the silencing of the lnc-uc.147 (Figure 1A).

The silencing of lnc-uc.147 reduced the cell viability (*p* < 0.001) and colony formation (*p* < 0.0001) of HepG2 cells compared to control cells (Figure 1B–D). 

No difference was seen on phenotypes such as proliferation, cell death, cell cycle, and migration after the knockdown of lnc-uc.147 compared to the control (Figure 2). 

Considering the biological role of lnc-uc.147 in BC and liver cancer cells, we aimed to identify how lnc-uc.147 could be regulated. Through ChiP-seq data from the UCSC Genome Browser, we verified that TEA domain transcription factor 4 (TEAD4) is able to bind to the uc.147 region (Figure 3A). To further verify this interaction in silico, the TFBSPred online tool was utilized, and binding sites for TEAD4 were observed through the uc.147 chromosome localization (Figure 3B). 

To validate the lnc-uc.147 regulation by TEAD4, the cell lines HepG2 from HCC, CAMA-1, and BT-474 cells, derived from BC, were transfected with a siRNA targeting the TEAD4 sequence. 

We obtained a knockdown of more than 50% of TEAD4 in all three cell lines 24 and 48 h after transfection (Figure 3C–E). We could also observe a decrease in the lnc-uc.147 expression 24 and 48 h after the transfection with siTEAD4 in CAMA-1 and BT-474 cell lines (Figure 3F–H). In HepG2 cells, a reduction in lnc-uc,147 expression was observed after 48 h. These results suggest that TEAD4 expression can regulate the levels of lnc-uc.147.

## 4. Discussion

In HepG2, an HCC cell line, we verified that silencing of lnc-uc.147 decreased the viability and clonogenic capacity. Those results were also observed in the BC cell lines, as Zambalde et al. (2021) described [5]. However, no alteration in the cell cycle and cell death was observed in the liver cells, which were altered in BC. 

Based on these results, we hypothesize that lnc-uc.147 is more related to pathways involved in the cell’s metabolism, not necessarily leading to cell death signaling. In addition, the increased gene expression of lnc-uc.147 was associated with more advanced disease stages when we have a high probability of metastasis. The colony assay distantly mimics the event of the tumor cell arriving at a new location in order to reactivate its proliferation mechanism without contact with neighboring tumor cells. In the proliferation assay, the major confluence allows cell communication. Therefore, the silencing of the gene affects this ability to colonize new tissues, which explains why its silencing only affected the formation of colonies and viability, and not their proliferation rate. The results described here demonstrate that lnc-uc.147 has a potential oncogenic role in liver cells as well, but the pathways involved in liver and breast models could be different.

The interaction between lnc-uc.147 and TEAD4 was suggested by an in silico analysis on the TFBSPred tool. The DNA binding domain of TEAD4 is reported to bind to the M-CAT (5′-CATTCCT-3′) regulatory element [13,14], which is present in the region of uc.147. Additionally, through an in vitro approach, we observed that after silencing TEAD4, the expression of lnc-uc.147 was diminished in breast and liver cells. Those results associated with the ChiP-Seq data from the UCSC database, previously showing a physical interaction between TEAD4 and uc.147, suggested the participation of this TF in lnc-uc.147 regulation, a mechanism that can be further explored in future studies. 

TEAD4 is a participant in the conserved signaling pathway HIPPO. This pathway is related to the control of tissue growth during development and is associated with pathologies such as cancer [15]. The expression of TEAD4 is increased in many types of cancer [16], including breast [17] and liver cancer [18,19]. Interestingly, in breast cancer, the complex between YAP1/TEAD4 can bind to enhancer regions of the estrogen receptor gene, causing an increase in the proliferation of estrogen-responsive breast cells [20]. We previously showed the high expression of lnc-uc.147 in BC luminal subtype [5], which indicates a potential regulation of this lncRNA by the TEAD4 transcription factor. 

Here, we demonstrate a relevant oncogenic role of the TEAD4 transcription factor in cancer, which correlates with the oncogenic role of lnc-uc.147—described in BC and herein in liver—suggesting that the overexpression of lnc-uc.147 could be induced by the TEAD4 transcription factor. 

This study suggests that lnc-uc.147 also has an important role beyond BC, including hepatocellular carcinoma, and potentially stomach adenocarcinoma and clear renal cell carcinoma. 

## 5. Conclusions

In conclusion, it was possible to observe the effect of lnc-uc.147 on liver cancer. We also suggested a possible regulation mechanism of this lncRNA by the TEAD4 transcription factor. Our results reinforce the importance of the transcribed ultraconserved regions and their emergence as potential targets in cancer.

## Figures and Tables

**Figure 1 biomolecules-13-00265-f001:**
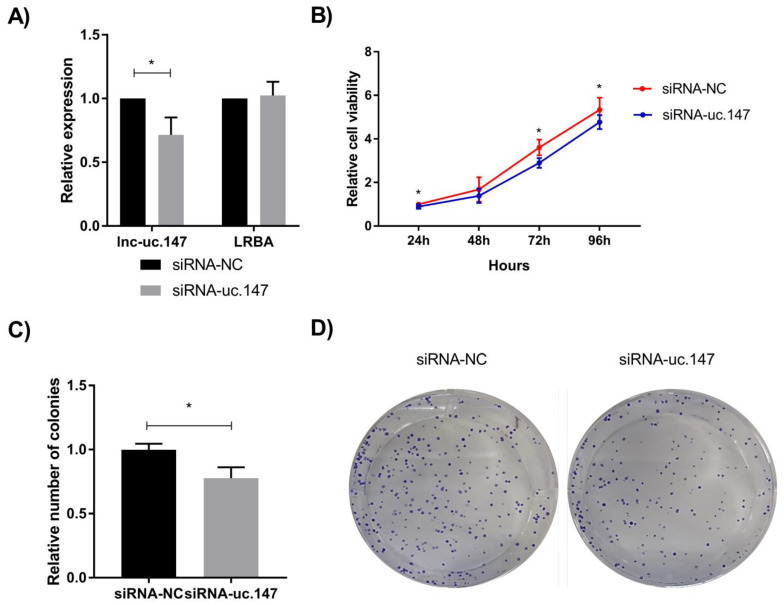
Cell viability and colony formation assays of lnc-uc.147 in HepG2 cells. (**A**) Relative expression levels of lnc-uc.147 and the host gene *LRBA* after treatment with a siRNA-uc.147 compared to negative control (siRNA-NC). Data normalized by the endogenous controls *GAPDH/U6* and the negative control. (**B**) Cell viability in HepG2 cells after treatment with siRNA uc.147 (blue line) and siRNA-NC (red line) at 24, 48, 72, and 96 h. (**C**) Relative number of HepG2 colonies after siRNA treatment. (**D**) Representative plates of colony formation. For all experiments, three independent experiments are represented. Statistical differences were evaluated using the Student *t*-test. * *p* < 0.05.

**Figure 2 biomolecules-13-00265-f002:**
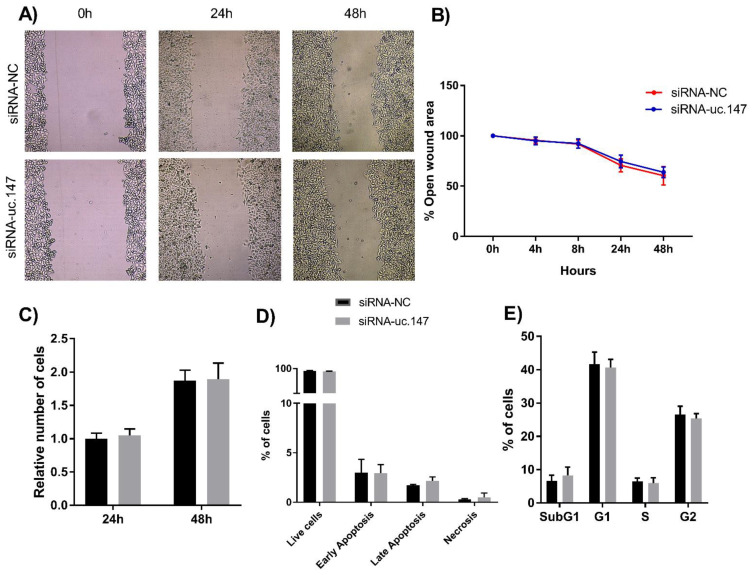
Migration, proliferation, apoptosis, and cell cycle assays. (**A**) Representative images of migration assay of HepG2 cells treated with siRNA-uc.147 and negative control at 0, 24, and 48 h after the scratch. (**B**) Percentage of open-wound area of migration assay at 0, 4, 8, 24, and 48 h. (**C**) Proliferation assay demonstrating the relative number of cells after 24 and 48 h. (**D**) Percentage of live, early apoptotic, late apoptotic, and necrotic cells after staining with Annexin-V/PI, and treatment with siRNAs. (**E**) Percentage of cells in the different stages of the cell cycle and SubG1 after staining with PI. For all experiments, three independent experiments are represented.

**Figure 3 biomolecules-13-00265-f003:**
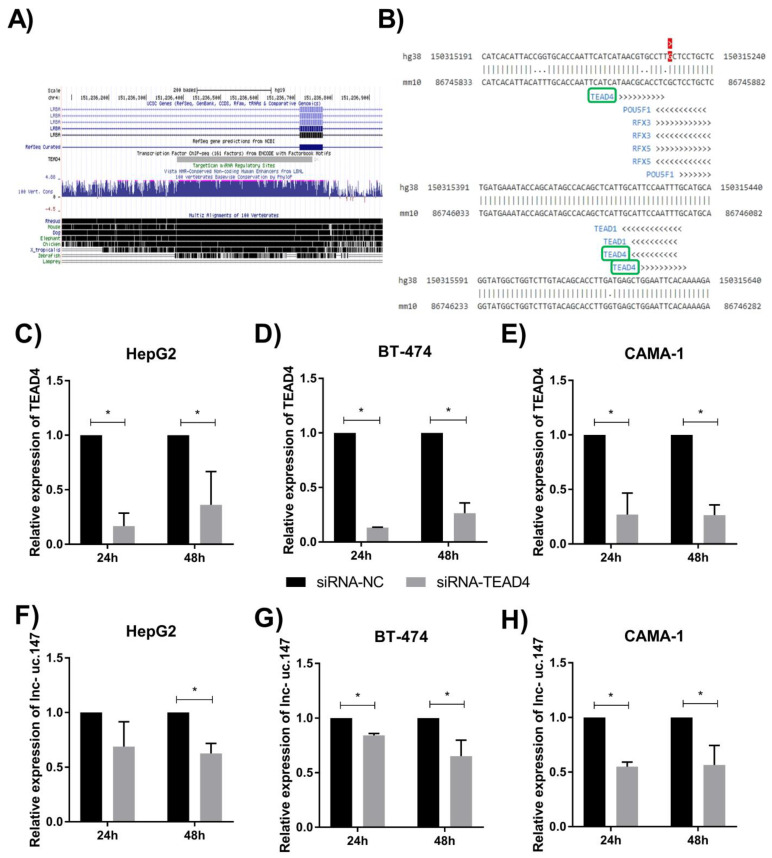
(**A**) UCSC Genome Browser Chip-Seq data showing that TEAD4 binds to the region of uc.147 (chr4:151236383-151236690—GRCh37). (**B**) Results from the TFBSPred tool demonstrate that, in the region of uc.147, binding sites for TEAD4 (marked in green) are predicted. For the analysis, the chromosome location of lnc-uc.147 (4:150315231(+)—GRCh38) was also used. RNA expression of the TEAD4 transcription factor after its silencing in the HepG2 (**C**), BT-474 (**D**), and CAMA-1 (**E**) cells. Expression of lnc-uc.147 after transcription factor silencing in HepG2 (**F**), BT-474 (**G**), and CAMA-1 (**H**) cells. Data normalized by the endogenous controls *GAPDH*/*U6*/*TBP* and the negative control. Three independent experiments are represented. Statistical differences were evaluated using the Student *t*-test. * *p* < 0.05.

**Table 1 biomolecules-13-00265-t001:** Primer sequence used in quantitative real-time PCR.

Genes		Sequence (5′-3′)
RNU6	Forward	CTCGCTTCGGCAGCACA
	Reverse	AACGCTTCACGAATTTGCGT
TBP	Forward	TCAAACCCAGAATTGTTCTCCTTAT
	Reverse	CCTGAATCCCTTTAGAATAGGGTAGA
GAPDH	Forward	GGATTTGGTCGTATTGGG
	Reverse	GGAAGATGGTGATGGGATT
uc.147	Forward	CGTCCTAGGCCTGTTCAAAT
	Reverse	TGGGAATGGAATTTTCCTGA
LRBA	Forward	CCAACTTCAGAGATTTGTCCAAGC
	Reverse	ATGCTGCTCTTTTTGGGTTCAG
TEAD4	Forward	GGACACTACTCTTACCGCATCC
	Reverse	TCAAAGACATAGGCAATGCACA

**Table 2 biomolecules-13-00265-t002:** TCGA analysis of the uc.147 region.

Type of Cancer	Association	*p*-Value
Breast invasive carcinoma (BRCA)	PAM50	2.72 × 10^−12^
	ER-positive	2.74 × 10^−6^
	Therapy	5.16 × 10^−2^
Liver hepatocellular carcinoma (LIHC)	Histologic grade	0.0181
Stomach adenocarcinoma (STAD)	Disease stage	0.0228
Kidney renal clear cell carcinoma (KIRC)	Disease stage	0.0489

Significant associations between tumor types and uc.147 expressions after analysis on the TANRIC platform. PAM50: a molecular study of the expression of 50 genes important for breast cancer. ER-positive: a tumor that expresses the estrogen receptor. A *p*-value < 0.05 was considered significant.

## Data Availability

The data that support the findings of this study are available in The Cancer Genome Atlas (TCGA) at https://www.tanric.org, reference number [6], accessed on 14 November 2021. These data were derived from the following resources available in the public domain: https://www.cancer.gov/about-nci/organization/ccg/research/structural-genomics/tcga, accessed on 14 November 2021.

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
