# Peer review of "Lnc-uc.147 Is Associated with Disease Stage of Liver, Gastric, and Renal Cancer"

_biomolecules, 2023, doi:10.3390/biom13020265_

Round 1
Reviewer 1 Report
In the present short report Rodrigues et al showed that Lnc-uc147 is dysregulated in hepatocellular, clear cell renal, and gastric adenocarcinoma. Silencing lnc-uc 147 reduces cell viability and clonogenic capacity in HepG2 cell lines. The transcription factor TEAD4 appears to regulate this lncRNA in liver and breast cancer cells. The study is interesting and I have several comment to be address before accepting this paper for publication biomolecules.
Comments:
Line 56: Define for which cancers these cell lines were used the first time you mentioned them: HepG2, BT-474, and CAMA-1 cell lines
The authors claim that Lnc-uc147 is dysregulated in liver, stomach and renal cancers but did not provide any data supporting this except those shown in table 2, the authors need to provide the figures dipecting this data e.g., association and also the authors showed that silencing Lnc-uc147 reduces its level that’s very expected but how can the readers including this reviewer be convinced that siRNA-NC (that’s cancer cell line) has a dysregulated level of Lnc-uc147 if no wild-type cell line of those tumors is used as control.
Line 71: change u6 to U6 and ug or uM should be written as µg or µM..
Table 2: change the decimal of p value from 0,0 to 0.0
Please provide the sequences of siRNA for TEAD4
If silencing Lnc-uc147 did reduce cell viability but has no effect on cell proliferation, cell apoptosis, and cell death, then what are the possible mechanisms that underpin the reduction in cell viability of Lnc-uc147??Describe them in your discussion
Since p-value< 0.05 was considered significant, there is no reason to show **, ***, **** for 0.01, 0.001, 0.0001 etc.
In figure 1a, why no standard error is shown in siRNA-NC and which statistical test was use in the Fig1a, and b and c? Describe that in the legend. Additionally the statistical analysis on Fig1b seems not accurate as the error bar are touching each other
In figure 3c-h why no standard error is shown in siRNA-NC?
The method 2.5 and 2.6 is not clear what’s the difference between them?
Methods should include a few references more
The paper will need to be moderately modified for the typos, grammar, and compositions
Author Response
According to reviewer 1:
“In the present short report Rodrigues et al showed that Lnc-uc147 is dysregulated in hepatocellular, clear cell renal, and gastric adenocarcinoma. Silencing lnc-uc 147 reduces cell viability and clonogenic capacity in HepG2 cell lines. The transcription factor TEAD4 appears to regulate this lncRNA in liver and breast cancer cells. The study is interesting and I have several comment to be address before accepting this paper for publication biomolecules.
Comments:
Line 56: Define for which cancers these cell lines were used the first time you mentioned them: HepG2, BT-474, and CAMA-1 cell lines”
Response: We agree with the suggestion, and we included the cancer origin of these cell lines in line 56, page 2 (highlighted in the manuscript).
“The authors claim that Lnc-uc147 is dysregulated in liver, stomach and renal cancers but did not provide any data supporting this except those shown in table 2, the authors need to provide the figures dipecting this data e.g., association”
Response: As this manuscript is a communication, we included the graphs that support these associations as Supplementary Figure 1, in a new document, “Supplemental materials”.
“and also the authors showed that silencing Lnc-uc147 reduces its level that’s very expected, but how can the readers including this reviewer be convinced that siRNA-NC (that’s cancer cell line) has a dysregulated level of Lnc-uc147 if no wild-type cell line of those tumors is used as control.
Response: We are grateful for the relevant comment, but we would like to reinforce that, in liver hepatocellular carcinoma, we found differential expression based on histologic grade and not between non-tumoral and tumor samples. In this case, we did not evaluate the expression of this lncRNA in a non-tumoral cell line because the difference is not predicted to be between the non-tumoral vs tumoral conditions.
Additionally, it is important to highlight that many TCGA samples (including non-tumor and tumor samples) have no lnc-uc147 expression detection. Considering that HepG2 cell line had lnc-uc147 expression detection, it was possible to compare the differences between HepG2 cells with more Lnc-uc147 expression or the same cell with less expression (after silencing). So, our report describes the effect of lnc-uc147 silencing in a cell line that originally has this lncRNA, independently of the deregulated level compared with other cell lines.
“Line 71: change u6 to U6 and ug or uM should be written as µg or µM.
Table 2: change the decimal of p value from 0,0 to 0.0”
Response: We thank you for your suggestions and corrected the main text (changes highlighted).
“Please provide the sequences of siRNA for TEAD4”
Response: We used predesigned TEAD4 siRNA from Sigma-Aldrich. All predesigned siRNA carries a company guarantee, but the sequence is not provided. The product number is NM_003213, siRNA ID: SASI_Hs01_00046458, sequence start: 1490.
The complete information available from the company was included in the manuscript (line 79).
“If silencing Lnc-uc147 did reduce cell viability but has no effect on cell proliferation, cell apoptosis, and cell death, then what are the possible mechanisms that underpin the reduction in cell viability of Lnc-uc147?? Describe them in your discussion
Response: We added some hypotheses of why we observed altered viability and not proliferation. It is present in the manuscript in the second paragraph of the discussion, starting at line 226 (the text is highlighted).
Since p-value< 0.05 was considered significant, there is no reason to show **, ***, **** for 0.01, 0.001, 0.0001 etc.
Response: We changed this in the figures and figure legends. Therefore, only * is shown to indicate a significant difference.
“In figure 1a, why no standard error is shown in siRNA-NC and which statistical test was use in the Fig1a, and b and c? Describe that in the legend. Additionally the statistical analysis on Fig1b seems not accurate as the error bar are touching each other
In figure 3c-h why no standard error is shown in siRNA-NC?”
Response: In figures 1a and 3c-h, the siRNA-NC has no standard error bar because the analysis used this control as a reference (value of 1). This type of normalization is currently used in siRNA analysis (for example: DOI:10.3892/ijo.2012.1691; https://doi.org/10.3390/ijms21239306; DOI: 10.26508/lsa.202000788).
The statistical test used in Fig1a-c was the unpaired Student t-test. That information was added to the legend.
Related to statistical analysis on Fig1b, we repeated the statistical tests and confirm this result.
“The method 2.5 and 2.6 is not clear what’s the difference between them?”
Response: The difference between both assays is the parameter that is being evaluated. In the cell viability assay, we added resazurin to the cells, and we measured the mitochondrial viability after treatment. On the other hand, for the proliferation assay, we utilize crystal violet, a nuclear dye, in order to infer how many cells are present in the well after the treatment. Although they seem to be similar assays, they can measure different things. We agree that this difference was not clear in the manuscript. Therefore, we included brief information in the methodology (Lines 88 and 98).
“Methods should include a few references more”
Response: References for the viability and proliferation assay were added to the main text (lines 88 and 98).
“The paper will need to be moderately modified for the typos, grammar, and compositions”
Response: We thank the reviewer for the careful reading of our manuscript. We did an extensive review for correct typos, grammar, and compositions.
Reviewer 2 Report
Very good article
Author Response
According to reviewer 2:
“Very good article”
Response: We thank the reviewer for the careful reading of our manuscript.
Reviewer 3 Report
Jaqueline Carvalho de Oliveira and colleagues by searching the TCGA database found altered expression of Lnc-uc.147 lncRNA in hepatocellular, clear cell renal, and gastric adenocarcinoma. Then, by silencing the expression of Lnc-uc.14 in HepG2 cells, showed a reduction of growth of the cells and suggested an oncogenic role for this lncRNA. Finally, claimed that Lnc-uc.14 can be regulated by TEAD4 transcription factor.
There are several concerns with the conclusion made from these data:
1. The data need to be checked on the liver and other tumor samples as well as the cell lines.
2. Authors claim that Lnc-uc.14 was regulated by the transcription factor TEAD4. Based on the results presented in this study it seems some exaggeration to make such a conclusion. Some direct experiments are needed to clearly show if Lnc-uc.14 can be regulated by TEAD4. Therefore the title is inappropriate as well.
3. "Clinical features" as used in the abstract have not been described in the manuscript, so "disease stage" was used in line 144.
Minor typos:
Line 73: 0,5 to 0.5
In Table 2: the p-value numbers should be corrected.
In table 2: PAM 50 to PAM50
Author Response
According to reviewer 3:
“Jaqueline Carvalho de Oliveira and colleagues by searching the TCGA database found altered expression of Lnc-uc.147 lncRNA in hepatocellular, clear cell renal, and gastric adenocarcinoma. Then, by silencing the expression of Lnc-uc.14 in HepG2 cells, showed a reduction of growth of the cells and suggested an oncogenic role for this lncRNA. Finally, claimed that Lnc-uc.14 can be regulated by TEAD4 transcription factor.
There are several concerns with the conclusion made from these data:
- The data need to be checked on the liver and other tumor samples as well as the cell lines.”
Response: We are grateful for the relevant comment, and we agree that should be interesting to check the differential expression on other samples and cell lines.
All our analyses included TCGA samples, one of the largest world cohorts of over 30 human tumors. Taking into account that TCGA is one of the most used datasets worldwide, many groups focused on creating facilities for TCGA analysis, such as the TANRIC website (from MD Anderson Cancer Center, https://ibl.mdanderson.org/tanric/_design/basic/query.html). This platform, which is focused on lncRNA analysis, includes the option to perform the expression analysis by the position of the gene in the chromosome.
The search for the position was essential in the present manuscript because lnc-uc147 is a new lncRNA, not yet cataloged in ensemble and other lncRNA catalogs, being extremely difficult to look for this expression in other datasets (including patient samples and cell lines). For this reason, we focused on the present manuscript only TCGA dataset and included cell lines expression that we had available in the laboratory.
We agree with the importance of analysis of lnc-uc.147 expression in other samples. But, we believe that the publication of our data in a “short communication” form, only with data of TCGA, is essential to reinforce the value of this lncRNA, which is extremely unknown, and must call attention to this molecule that must be included in lncRNAs catalogs soon, so that could be better studied in the future.
“2. Authors claim that Lnc-uc.14 was regulated by the transcription factor TEAD4. Based on the results presented in this study it seems some exaggeration to make such a conclusion. Some direct experiments are needed to clearly show if Lnc-uc.14 can be regulated by TEAD4. Therefore the title is inappropriate as well.
Response: Considering that we had ChiP-Seq (data available from ENCODE project), prediction of binding sites for TEAD4 in the lnc-uc.147 sequence, and we showed altered expression of lncRNA after the modulation of TEAD4, we considered that we had enough evidence to suggest that this TF could regulate the lncRNA.
But we completely agree with the reviewer that TEAD4 regulation of lnc-uc147 is still a suggestion. So, we changed the manuscript title and revised the manuscript to be less emphatic and consider this regulation as a based evidence suggestion.
“3. "Clinical features" as used in the abstract have not been described in the manuscript, so "disease stage" was used in line 144.
Minor typos:
Line 73: 0,5 to 0.5
In Table 2: the p-value numbers should be corrected.
In table 2: PAM 50 to PAM50”
Response: We changed the “clinical features” expression to “disease stage”, considering that this was the association observed in the analysis.
All minor suggestions were accepted (changes highlighted), and we really thank the reviewer for the constructive comments and suggestions with respect to our manuscript.
Round 2
Reviewer 1 Report
The authors have addressed all my pointed nicely. This short communication may be accepted for publication in biomolecules
Reviewer 3 Report
Still minor English checking is required!